# Higher ratio of plasma omega-6/omega-3 fatty acids is associated with greater risk of all-cause, cancer, and cardiovascular mortality: A population-based cohort study in UK Biobank

Yuchen Zhang[1], Yitang Sun[2], Qi Yu[3], Suhang Song[4], J Thomas Brenna[5,6], Ye Shen[1]*, Kaixiong Ye[2,7]*

[1]Department of Epidemiology and Biostatistics, College of Public Health, University of Georgia, Athens, Georgia, United States; [2]Department of Genetics, University of Georgia, Athens, Georgia, United States; [3]Department of Biostatistics and Bioinformatics, Emory University, Atlanta, Georgia, United States; [4]Department of Health Policy and Management, College of Public Health, University of Georgia, Athens, Georgia, United States; [5]Division of Nutritional Sciences, Cornell University, Ithaca, New York, United States; [6]Dell Pediatric Research Institute and the Depts of Pediatrics, of Nutrition, and of Chemistry, University of Texas at Austin, Austin, Texas, United States; [7]Institute of Bioinformatics, University of Georgia, Athens, Georgia, United States

*For correspondence:
yeshen@uga.edu (YS);
kaixiong.ye@uga.edu (KY)

Competing interest: The authors declare that no competing interests exist.

## Abstract

**Background:** Circulating omega-3 and omega-6 polyunsaturated fatty acids (PUFAs) have been associated with various chronic diseases and mortality, but results are conflicting. Few studies examined the role of omega-6/omega-3 ratio in mortality.

**Methods:** We investigated plasma omega-3 and omega-6 PUFAs and their ratio in relation to all-cause and cause-specific mortality in a large prospective cohort, the UK Biobank. Of 85,425 participants who had complete information on circulating PUFAs, 6461 died during follow-up, including 2794 from cancer and 1668 from cardiovascular disease (CVD). Associations were estimated by multivariable Cox proportional hazards regression with adjustment for relevant risk factors.

**Results:** Risk for all three mortality outcomes increased as the ratio of omega-6/omega-3 PUFAs increased (all $P_{trend}$ <0.05). Comparing the highest to the lowest quintiles, individuals had 26% (95% CI, 15–38%) higher total mortality, 14% (95% CI, 0–31%) higher cancer mortality, and 31% (95% CI, 10–55%) higher CVD mortality. Moreover, omega-3 and omega-6 PUFAs in plasma were all inversely associated with all-cause, cancer, and CVD mortality, with omega-3 showing stronger effects.

**Conclusions:** Using a population-based cohort in UK Biobank, our study revealed a strong association between the ratio of circulating omega-6/omega-3 PUFAs and the risk of all-cause, cancer, and CVD mortality.

**Funding:** Research reported in this publication was supported by the National Institute of General Medical Sciences of the National Institute of Health under the award number R35GM143060 (KY). The content is solely the responsibility of the authors and does not necessarily represent the official views of the National Institutes of Health.

## eLife assessment

The manuscript provides **convincing** evidence that both circulating omega-6 and omega-3 PUFAs are associated with lower all-cause, cancer, and cardiovascular mortality in the UK BioBank population and that omega-3s have a stronger effect than omega-6s. The findings have **important** public health implications.

## Introduction

Cancer and cardiovascular disease (CVD) are the two leading causes of non-communicable disease mortality globally (*Vos et al., 2020*). Substantial epidemiologic evidence has linked the dietary or circulating levels of omega-3 and omega-6 polyunsaturated fatty acids (PUFAs) to the risk of all-cause, cancer, and CVD mortality (*Supplementary file 1* Table S1). However, the results are inconsistent, especially for omega-6 PUFAs. Most large observational studies support the inverse associations between omega-3 PUFAs and mortality. In addition to total and individual omega-3 PUFAs, the omega-3 index, defined as the percentage of eicosapentaenoic acid (EPA) and docosahexaenoic acid (DHA) in total fatty acids in red blood cells, was shown to be a validated biomarker of the dietary intake and tissue levels of long-chain omega-3 PUFAs, and was proposed to be a risk factor for CVD and related mortality (*Harris, 2009*; *Harris and Von Schacky, 2004*). While the omega-3 index has been observed to be inversely associated with all cause-mortality, its association patterns with CVD and cancer mortality are less clear (*Harris et al., 2017*; *Harris et al., 2018*; *Kleber et al., 2016*). Most importantly, results from clinical trials of omega-3 PUFA supplementation have been inconsistent (*Bhatt et al., 2019*; *Nicholls et al., 2020*). On the other hand, while some studies revealed inverse associations of omega-6 PUFAs with all-cause mortality (*Delgado et al., 2017*; *Marklund et al., 2019*; *Wu et al., 2014*), others reported null results (*Harris et al., 2017*; *Harris et al., 2020*; *Miura et al., 2016*; *Otsuka et al., 2019*; *Zhuang et al., 2019*). The role of omega-6 PUFAs in cancer and CVD mortality is less studied, and the patterns are similarly conflicting (*Harris et al., 2017*; *Delgado et al., 2017*; *Marklund et al., 2019*; *Wu et al., 2014*). Interpreting previous observational results is challenging due to the limitations of small sample sizes, insufficient adjustments for confounding, and unique sample characteristics. Moreover, many studies rely on self-reported dietary intake or fish oil supplementation status, which are subject to large variability and reporting bias (*Shim et al., 2014*). Despite the tremendous interest and research effort, the roles of omega-3 and omega-6 PUFAs in all-cause and cause-specific mortality remain uncertain.

It has been suggested that the high omega-6/omega-3 ratio in Western diets, 20:1 or even higher, as compared to an estimated 1:1 during the most time of human evolution, contributes to many chronic diseases, including CVD, cancer, and autoimmune disorders (*Simopoulos, 2001*; *Simopoulos, 2008*). However, while many previous studies have examined total or individual omega-3 and omega-6 PUFAs, fewer investigated the role of their imbalance, as measured by the omega-6/omega-3 ratio, in mortality (*Harris et al., 2017*; *Otsuka et al., 2019*; *Zhuang et al., 2019*; *Noori et al., 2011*). In a prospective cohort study of postmenopausal women, the omega-6/omega-3 ratio in red blood cells was associated with an increased risk of all-cause mortality but not with cancer or CVD mortality (*Harris et al., 2017*). Similar positive associations with all-cause mortality were observed in smaller cohorts investigating the ratio in serum or dietary intake (*Otsuka et al., 2019*; *Noori et al., 2011*). However, prospective studies in two independent cohorts from China and US did not find a linear positive association between the omega-6/omega-3 ratio in diet and all-cause mortality (*Zhuang et al., 2019*). To address these gaps in our understanding of the roles of omega-3 and omega-6 PUFAs and their imbalance in all-cause and cause-specific mortality, we perform a prospective study in a large population-based cohort (N=85,425) from UK Biobank, using objective measurements of PUFA levels in plasma.

**eLife digest** Fatty acids play an essential role in health. Studies have shown that diets high in omega-3 fatty acids found in foods like fish, fish oil, flaxseed and walnuts may be beneficial. Yet some studies have raised concern that too many omega-6 fatty acids in Western diets rich in vegetable oils may be harmful. Some scientists have proposed that the balance of omega-3 and omega-6 in diets is vital to health. They hypothesize that a higher omega-6 to omega-3 fatty acids ratio is detrimental.

But, proving that a higher ratio of omega-6 to omega-3 fatty acids is harmful has been difficult. Many studies have found conflicting results. Scientists have struggled to accurately measure fatty acid intake as tracking an individual's dietary intake is challenging and self-reported dietary intake may be incorrect. Additionally, scientists must follow individuals for many years to determine if a high ratio of omega-6 to omega-3 is linked with cancer, heart disease, or death. But, measuring circulating fatty acids in an individual's blood may offer an easier and more reliable approach to studying the health impacts of these vital nutrients.

Zhang et al. show that people with higher ratios of omega-6 to omega-3 fatty acids in their blood are at greater risk of dying from cancer, heart disease, or any cause than those with lower ratios. The experiments measured omega-6 and omega-3 fatty acid levels in more than 85,000 participants in the UK Biobank who scientists followed for an average of about 13 years. Participants with the highest ratios of omega-6 to omega-3 fatty acids were 26% more likely to die of any cause, 14% more likely to die of cancer, and 31% more likely to die of heart disease than individuals with the lowest ratios. Individually, high levels of omega-6 fatty acids and high levels of omega-3 fatty acids were both associated with a lower risk of dying. But the protective effects of omega-3 were greater. For example, individuals with the highest levels of omega-6 fatty acids were 23% less likely to die of any cause. By comparison, those with the highest levels of omega-3s were 31% less likely to die. The stronger protection offered by high levels of omega-3s likely explains why having a high ratio of omega-6s to omega-3s was linked to harm. Both are protective. But the protection provided by omega-3s is more robust.

The experiments support dietary interventions to raise omega-3 fatty acid levels and maintain a low omega-6 to omega-3 fatty acid ratio to prevent early deaths from cancer, heart disease or other causes. More research is needed to understand the impact of dietary fatty acid intake on other diseases and how genetics may influence the health impact of fatty acids.

## Methods
### Study population
The UK Biobank study is a prospective, population-based cohort study in the United Kingdom (*Sudlow et al., 2015*). Between 2006 and 2010, 502,384 prospective participants, aged 40–69, in 22 assessment Centers throughout the UK were recruited for the study. The population information was collected through a self-completed touch-screen questionnaire; brief computer-assistant interview; physical and functional measures; and blood, urine, and saliva collection during the assessment visit. Participants with cancer (n=37,736) or CVD (n=100,972), those who withdrew from the study (n=879), and those with incomplete data on the plasma omega-6/omega-3 ratio (n=277,372) were excluded from this study, leaving 85,425 participants, 6461 died during follow-up, including 2794 from cancer and 1668 from CVD.

### Ascertainment of exposure
Metabolomic profiling of plasma samples was performed with high-throughput nuclear magnetic resonance (NMR) spectroscopy. At the time of this analysis (15 Mar 2023), UK Biobank released the Phase 1 metabolomic dataset, which covered a random selection of 118,461 plasma samples from the baseline recruitment. These samples were collected between 2007 and 2010 and had been stored in −80 °C freezers, while the NMR measurements took place between 2019 and 2020. Detailed descriptions can be found in previous publications about plasma sample preparation, NMR spectroscopy setup, quality control protocols, correction for sample dilution, verification with duplicate samples and internal controls, and comparisons with independent measurements from clinical chemistry assays

(*Sudlow et al., 2015*; *Julkunen et al., 2023*; *Würtz et al., 2017*). Five PUFAs-related biomarkers were directly measured in absolute concentration units (mmol/L), including total PUFAs, total omega-3 PUFAs, total omega-6 PUFAs, docosahexaenoic acid (DHA), and linoleic acid (LA). Of note, DHA is one type of omega-3 PUFAs, and LA is one type of omega-6 PUFAs. Our primary exposure of interest, the omega-6/omega-3 ratio, was calculated based on their absolute concentrations. We also performed supplemental analysis for four exposures, the percentages of omega-3 PUFAs, omega-6 PUFAs, DHA, and LA in total fatty acids (omega-3%, omega-6%, DHA%, and LA%), which were calculated by dividing their absolute concentrations by that of total fatty acids.

## Ascertainment of outcome

The date and cause of death were identified through the death registries of the National Health Service (NHS) Information Centre for participants from England and Wales and the NHS Centre Register Scotland for participants from Scotland (*Sudlow et al., 2015*). At the time of the analysis (March 15, 2023), we had access to the most up-to-date mortality dataset (Version: December 2022), which had the date of death up to 12 November 2021. Therefore, follow-up time was calculated as the time between the date of entering the assessment centre and this date, or the date of death, whichever happened first. The underlying cause of death was assigned and coded in vital registries according to the International Classification of Diseases, 10th revision (ICD-10). CVD mortality was defined using codes I00-I99, and cancer mortality was defined using codes C00-D48. To exclude those who had cancer at baseline, we used the overall cancer incidence information based on diagnoses in cancer registers derived from the Health and Social Care Information Centre and the NHS Central Register that were coded based on ICD-9 and ICD-10 codes. The information about CVD incidence was retrieved from Hospital inpatient data and Death Register records that were coded according to ICD-9 and ICD-10 codes, and self-reported medical condition.

## Ascertainment of covariates

The baseline questionnaire included detailed information on several possible confounding variables: demographic factors (age, sex, assessment centre, ethnicity), socioeconomic status (Townsend Deprivation Index), lifestyle habits (alcohol assumption, smoking status, body mass index (BMI), physical activity, comorbidities (including hypertension, diabetes, and longstanding illness)), and other supplementation (fish-oil supplementation). The Townsend Deprivation Index, used as an indicator of socioeconomic status, is retrieved directly from the UK Biobank. BMI was defined as the body mass divided by the square of the body height and was expressed in units of kg/m². Comorbidities, including hypertension, diabetes, and longstanding illness, were self-reported at baseline. Longstanding illness refers to any long-standing illness, disability, or infirmity, without other specific information. We also retrieved the information on dietary PUFA intakes and serum biomarkers for secondary analysis. Dietary PUFAs were estimated based on the 24 hr dietary recall (*Perez-Cornago et al., 2021*). The percentage of dietary omega-3 and omega-6 PUFAs were calculated as the corresponding absolute dietary PUFAs divided by summation of dietary monounsaturated fatty acids, saturated fatty acids, omega-3 fatty acids, and omega-6 fatty acids. The biochemistry markers were measured in the blood samples collected at recruitment.

## Statistical analysis

We summarized and compared the characteristics of the participants across quintiles of the omega-6/omega-3 ratio at baseline using descriptive statistics. Pearson's Chi-squared test and ANOVA test were used to compare the demographic characteristics across quintiles, respectively, for categorical variables and continuous variables. To investigate associations of the ratio with all-cause and cause-specific mortality, we used multivariable Cox proportional hazards regression models to calculate hazard ratios (*Chua et al., 2012*) and their 95% confidence intervals (CI). The proportional hazards assumption was not violated based on Schoenfeld residuals. We analyzed the ratio as continuous and categorical variables (i.e. quintiles). For all trend tests, we used the median value of each quintile as a continuous variable in the models. Potential nonlinear associations were assessed semi-parametrically using restricted cubic splines (4 knots were used in regression splines; *Durrleman and Simon, 1989*).

Based on previous literature and biological plausibility (*Wu et al., 2014*; *Li et al., 2020*; *Zhang et al., 2018*), we chose the following variables as covariates in the multivariable models: age (years;

continuous), sex (male, female), race (White, Black, Asian, Others), Townsend deprivation index (continuous), assessment centre, BMI (kg/m$^2$; continuous), smoking status (never, previous, current), alcohol intake status (never, previous, current), physical activity (low, moderate, high), and comorbidities (yes, no).

In secondary analyses to assess potential differences in associations across different population subgroups, we repeated the above-described analyses stratified by age (<vs.≥the median age of 58 years), sex (male/female), Townsend deprivation index (<vs.≥the population median of –2), BMI (<vs.≥25), comorbidities (yes vs. no), physical activity (low and moderate vs. high) and current smoking status (yes vs. no). Besides, to compare the effects of dietary intake of PUFAs with the plasma levels on mortality, we repeated the above-described analyses in participants with complete information on dietary intake PUFAs (N=153,064). Moreover, we conducted mediation analysis to investigate potential biological pathways that might explain the effects of omega-6/omega-3 ratio on mortality outcomes. Based on plausibility, we chose CRP as a potential mediating factor for CVD mortality; SHBG, TTST, E2 and IGF-1 as potential mediating factors for cancer mortality; and all the above biomarkers for all-cause mortality. We first examined the variations in the concentrations of each biomarker according to omega-6/omega-3 ratio quintiles. Then we constructed two regression models, a linear model, to regress the mediator (treated the biomarkers as continuous) on the exposure (treated the omega-6/omega-3 ratio as continuous), with fully adjustment for potential confounders; and a Cox proportional hazards model, to regress the outcome (mortality outcomes) on the exposure and the mediator, with fully-adjustment for potential confounders. We chose to construct Cox models for mediation analysis because our outcomes were rare (all <10%; *Lapointe-Shaw et al., 2018*). We integrated these two regression models to obtain the indirect effects and proportions of mediating effects through the counterfactual-based approach proposed by Vander Weele (*Valeri and VanderWeele, 2015*; *Vander-Weele, 2011*).

We also conducted several sensitivity analyses. First, to assess whether the associations of the omega-6/omega-3 ratio with mortality outcomes were primarily driven by omega-3 fatty acids or omega-6 fatty acids, we assessed both the separate and the joint associations of omega-3 fatty acids to total fatty acids percentage (omega-3%) and omega-6 fatty acids to total fatty acids percentage (omega-6%) with the three mortality outcomes. We also performed a joint analysis with categories of the omega-3% and omega-6% quintiles, using participants in both the lowest omega-3% and omega-6% quintiles as the reference category. An interaction term between omega-3% and omega-6% was included in the multivariable Cox proportional hazards model, and a Likelihood Ratio test was used to assess its significance. The correlation between omega-3% and omega-6% was assessed by the Pearson correlation. Second, to assess the effects of individual PUFAs on mortality, we conducted the same analysis for circulating docosahexaenoic acid to total fatty acids percentage (DHA%) and linoleic acid to total fatty acids percentage (LA%) with the three mortality outcomes. Third, to address the potential residual confounding by fish oil supplementation, we further adjusted for the supplementation status from the baseline questionnaire. Fourth, to investigate the effects of missing values, we imputed missing values (<1% for most factors, up to 19% for physical activity) by chained equations and performed sensitivity analyses on the imputed datasets (*Groothuis-Oudshoorn, 2011*). Fifth, to assess whether the observed associations are attenuated by reverse causation, we excluded those who died in the first year of follow-up. Last, to assess the representativeness of the participants included in our study, we compared the baseline characteristics between the participants with and without exposure information. All p-values were two-sided. We considered a p-value <0.05 or a 95% confidence interval (CI) excluding 1.0 for HRs as statistically significant. We conducted all analyses using R, version 4.0.3 and SAS statistical software, version 9.4 (SAS Institute Inc).

## Results

### Baseline characteristics

In the analytic cohort of 85,425 participants, over a mean of 12.7 years of follow-up, 6461 died, including 2794 from cancer and 1668 from CVD. The baseline characteristics of the participants across quintiles of the ratio of omega-6/omega-3 were summarized in *Table 1*. Study participants were, on average, 56 years old and 90% White. Those in the higher ratio quintiles were more likely

**Table 1.** Selected participaneline across quintiles of the plasma omega-6/omega-3 PUFAs ratio (n=85,425).

| Characteristics* | Omega-6/omega-3 ratio quintiles | | | | | |
|---|---|---|---|---|---|---|
| | 1 (median = 5.9) (n=17,085) | 2 (median = 7.6) (n=17,085) | 3 (median = 9.1) (n=17,085) | 4 (median = 11.0) (n=17,085) | 5 (median = 14.8) (n=17,085) | p |
| Age (years) | 58.6 (7.5) | 57.2 (7.9) | 55.7 (8.2) | 54.7 (8.3) | 53.4 (8.2) | <0.001† |
| Sex (male%) | 40.3 | 44.9 | 47.5 | 49.4 | 52.8 | <0.001‡ |
| Ethnicity(n%) | | | | | | |
| White | 15,375 (90.4%) | 15,486 (91.0%) | 15,494 (91.1%) | 15,467 (91.1%) | 15,420 (90.7%) | 0.384‡ |
| Black | 113 (0.7%) | 99 (0.6%) | 105 (0.6%) | 92 (0.5%) | 116 (0.7%) | |
| Asian | 673 (4.0%) | 639 (3.8%) | 663 (3.9%) | 653 (3.8%) | 659 (3.9%) | |
| Others | 844 (5.0%) | 786 (4.6%) | 742 (4.4%) | 773 (4.6%) | 810 (4.8%) | |
| *Missing (n)* | *80* | *75* | *81* | *100* | *80* | |
| BMI | 27.0 (4.4) | 27.3 (4.5) | 27.3 (4.7) | 27.3 (4.8) | 26.9 (5.0) | <0.001† |
| *Missing (n)* | *54* | *64* | *55* | *58* | *71* | |
| TDI | –1.6 (3.0) | –1.5 (3.0) | –1.4 (3.1) | –1.2 (3.1) | –0.9 (3.2) | <0.001† |
| *Missing (n)* | *17* | *24* | *24* | *28* | *22* | |
| Smoking status (n%) | | | | | | <0.001‡ |
| Never | 9426 (55.5%) | 9272 (54.6%) | 9424 (55.4%) | 9400 (55.3%) | 9214 (54.2%) | |
| Previous | 6402 (37.7%) | 6194 (36.5%) | 5772 (34.0%) | 5492 (32.3%) | 4951 (29.1%) | |
| Current | 1,162 (6.8%) | 1,526 (9.0%) | 1,802 (10.6%) | 2,104 (12.4%) | 2,832 (16.7%) | |
| *Missing (n)* | *95* | *93* | *87* | *89* | *88* | |
| Alcohol status (n%) | | | | | | <0.001‡ |
| Never | 631 (3.7%) | 676 (4.0%) | 655 (3.8%) | 755 (4.4%) | 974 (5.7%) | |
| Previous | 536 (3.1%) | 554 (3.2%) | 535 (3.1%) | 617 (3.6%) | 835 (4.9%) | |
| Current | 15,877 (93.2%) | 15,822 (92.8%) | 15,847 (93.0%) | 15,664 (91.9%) | 15,221 (89.4%) | |
| *Missing (n)* | *41* | *33* | *48* | *49* | *55* | |
| Physical activity (n%) | | | | | | <0.001‡ |
| Low | 2422 (17.4%) | 2602 (18.8%) | 2653 (19.1%) | 2617 (19.0%) | 2613 (18.9%) | |
| Moderate | 5879 (42.2%) | 5762 (41.5%) | 5712 (41.1%) | 5533 (40.1%) | 5327 (38.6%) | |
| High | 5635 (40.4%) | 5506 (39.7%) | 5545 (39.9%) | 5654 (41.0%) | 5854 (42.4%) | |
| *Missing (n)* | *3149* | *3215* | *3175* | *3281* | *3291* | |
| Fish oil supplementation (Yes%) | 48.1 | 37.7 | 29.6 | 22.7 | 15.5 | <0.001‡ |
| *Missing (n)* | *50* | *47* | *77* | *82* | *71* | |
| Comorbidity (Yes%) | 38.6 | 37.3 | 34.2 | 33.8 | 33.0 | <0.001‡ |
| Plasma omega-3 percentage | 6.6 (1.4) | 4.9 (0.5) | 4.2 (0.4) | 3.5 (0.3) | 2.6 (0.5) | <0.001† |
| Plasma omega-6 percentage | 36.3 (3.7) | 37.4 (3.4) | 38.2 (3.3) | 39.1 (3.2) | 40.2 (3.1) | <0.001† |
| Plasma DHA percentage | 2.8 (0.8) | 2.1 (0.5) | 1.9 (0.4) | 1.7 (0.4) | 1.5 (0.4) | <0.001† |
| Plasma LA percentage | 27.0 (3.3) | 28.4 (3.1) | 29.3 (3.0) | 30.2 (3.0) | 31.3 (3.1) | <0.001† |
| Dietary omega-3 percentage | 3.6 (1.8) | 3.2 (1.4) | 3.0 (1.3) | 2.9 (1.2) | 2.7 (1.0) | <0.001† |
| *Missing (n)* | *9355* | *9628* | *9857* | *9974* | *10,288* | |
| Dietary omaga-6 percentage | 16.8 (4.6) | 16.5 (4.4) | 16.6 (4.6) | 16.5 (4.7) | 16.5 (4.8) | <0.001† |

*Table 1 continued on next page*

Table 1 continued

| | | | | | |
|---|---|---|---|---|---|
| Missing (n) | 9355 | 9628 | 9857 | 9974 | 10,288 |

*All variables measured at baseline are presented as mean (SD) unless otherwise specified.

†From the ANOVA test for continuous variables.

‡From the Pearson's Chi-squared test for categorical variables.

to be younger, male, and current smokers, but less likely to have comorbidities and take fish oil supplementation.

## Main results

The associations of the omega-6/omega-3 ratio with all-cause and cause-specific mortality were presented in *Table 2*. A higher ratio was strongly associated with higher mortality from all causes, cancer, and CVD ($P_{trend}$ <0.05 for all three). In the fully adjusted models that considered the ratio as a continuous variable, every unit increase in the ratio corresponded to 2%, 1%, and 2% higher risk in all-cause, cancer, and CVD mortality, respectively. When comparisons were made between the highest and the lowest quintile of the omega-6/omega-3 ratio, there were 26%, 14%, and 31% increased risk for all-cause, cancer, and CVD mortality, respectively.

## Stratified analysis

The fully adjusted associations of the omega-6/omega-3 ratio with all-cause mortality revealed that compared to the lowest quintile, the highest quintile has strong, statistically significant associations with elevated risk within all categories of age, sex, TDI, BMI, comorbidities, physical activity, and smoking status (*Figure 1*, *Supplementary file 2*: Table S2), except in those aged less than 58 years old and BMI less than 25. The estimated associations with all-cause mortality were stronger in current smokers (P for interaction <0.01; *Figure 1*, *Supplementary file 2*). For cancer and CVD mortality, they also tended to be stronger among current smokers but did not reach statistical significance (P for interaction = 0.13 and 0.12, respectively). The associations with CVD mortality tended to be stronger

**Table 2.** Associations* of the plasma omega-6/omega-3 PUFAs ratio with all-cause, cancer, and CVD mortality risk in the UK Biobank.

| | Causes of death | | | | | | | | |
|---|---|---|---|---|---|---|---|---|---|
| | All-cause | | | Cancer | | | Cardiovascular diseases | | |
| | Number of deaths | Partially adjusted associations† | Fully adjusted associations‡ | Number of deaths | Partially adjusted associations† | Fully adjusted associations‡ | Number of deaths | Partially adjusted associations† | Fully adjusted associations‡ |
| Omega ratio variable forms | | HR (95%CI) | HR (95%CI) | | HR (95%CI) | HR (95%CI) | | HR (95%CI) | HR (95%CI) |
| Continuous | 6461 | 1.02 (1.02–1.03) | 1.02 (1.02–1.03) | 2794 | 1.02 (1.01–1.03) | 1.01 (1.00–1.02) | 1668 | 1.02 (1.01–1.03) | 1.02 (1.01–1.03) |
| Quintiles (median) | | | | | | | | | |
| 1 (5.9) | 1348 | 1.00 (ref) | 1.00 (ref) | 593 | 1.00 (ref) | 1.00 (ref) | 369 | 1.00 (ref) | 1.00 (ref) |
| 2 (7.6) | 1256 | 1.00 (0.92–1.08) | 0.96 (0.88–1.05) | 563 | 1.02 (0.91–1.15) | 0.98 (0.86–1.12) | 315 | 0.90 (0.77–1.05) | 0.89 (0.75–1.06) |
| 3 (9.1) | 1236 | 1.06 (0.98–1.15) | 1.01 (0.93–1.11) | 543 | 1.08 (0.96–1.21) | 0.99 (0.87–1.13) | 321 | 0.97 (0.84–1.13) | 0.97 (0.81–1.16) |
| 4 (11.0) | 1252 | 1.14 (1.06–1.23) | 1.09 (1.00–1.19) | 548 | 1.16 (1.03–1.31) | 1.12 (0.98–1.27) | 306 | 0.98 (0.84–1.15) | 1.02 (0.86–1.22) |
| 5 (14.8) | 1369 | 1.34 (1.24–1.45) | 1.26 (1.15–1.38) | 547 | 1.27 (1.12–1.43) | 1.14 (1.00–1.31) | 357 | 1.20 (1.04–1.40) | 1.31 (1.10–1.55) |
| $P_{trend}$ | | <0.001 | <0.001 | | <0.001 | 0.011 | | 0.002 | <0.001 |

*From Cox proportional hazards regression.

†Adjusted for age (years; continuous), sex (male, female), race (White, Black, Asian, Others), Townsend deprivation index (continuous), assessment centre.

‡Adjusted for age (years; continuous), sex (male, female), race (White, Black, Asian, Others), Townsend deprivation index (continuous), assessment centre, BMI (kg/m2; continuous), smoking status (never, previous, current), alcohol intake status (never, previous, current), physical activity (low, moderate, high), and comorbidities (yes, no).

CI = confidence interval. HR = hazards ratio. ref = reference.

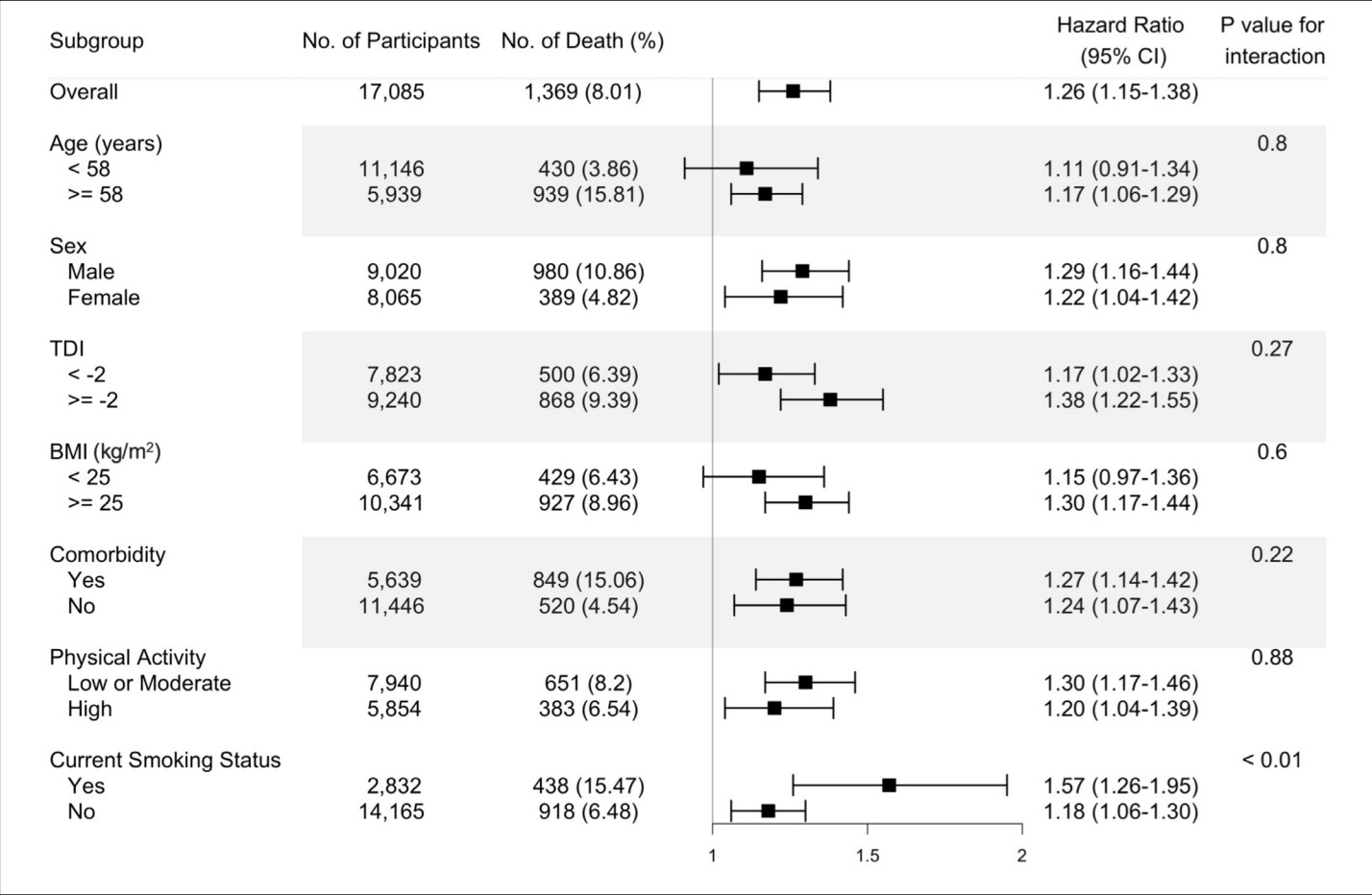

**Figure 1.** Risk estimates of all-cause mortality for the highest compared with the lowest quintile of the ratio of plasma omega-6 to omega-3 PUFAs, stratified by potential risk factors. Results were adjusted for age (years; continuous), sex (male, female), race (White, Black, Asian, Others), Townsend deprivation index (continuous), assessment centre, BMI (kg/m²; continuous), smoking status (never, previous, current), alcohol intake status (never, previous, current), physical activity (low, moderate, high), and comorbidities (yes, no).

among participants without comorbidities (P for interaction <0.01). No significant interactions were found for other risk factors (*Figure 2*).

## Restricted cubic spline analysis

Restricted cubic spline analysis suggested significant positive associations of the omega-6/omega-3 ratio with all-cause, cancer, and CVD mortality (p<0.05 for all three outcomes, *Figure 3*). Potential nonlinearity in these positive associations was identified for all-cause mortality (p<0.05) and CVD mortality (p<0.05) but not for cancer mortality (p=0.12). The strength of the relationship between the ratio and all-cause mortality appears to remain at a relatively low level before it starts to increase quickly after the ratio exceeds 8. A similar trend with higher uncertainties was observed for CVD mortality.

## Omega-3, Omega-6, Joint analysis, and comparison with dietary PUFAs

We further performed analyses to assess whether the associations of the omega-6/omega-3 ratio with mortality outcomes were primarily driven by omega-3 or omega-6 fatty acids. The correlation between omega-3% and omega-6% was relatively low with r=–0.12 (p<0.01). Across all models, both the omega-3% and omega-6% were inversely associated with all three mortality outcomes, except for plasma omega-6% with CVD mortality under Model 3 ($P_{trend}$ <0.01, *Supplementary file 2*: Tables S3 and S4). Notably, their associations remained significant when they were included in the same models. On the other hand, the effect sizes of the inverse associations were always bigger for the omega-3%

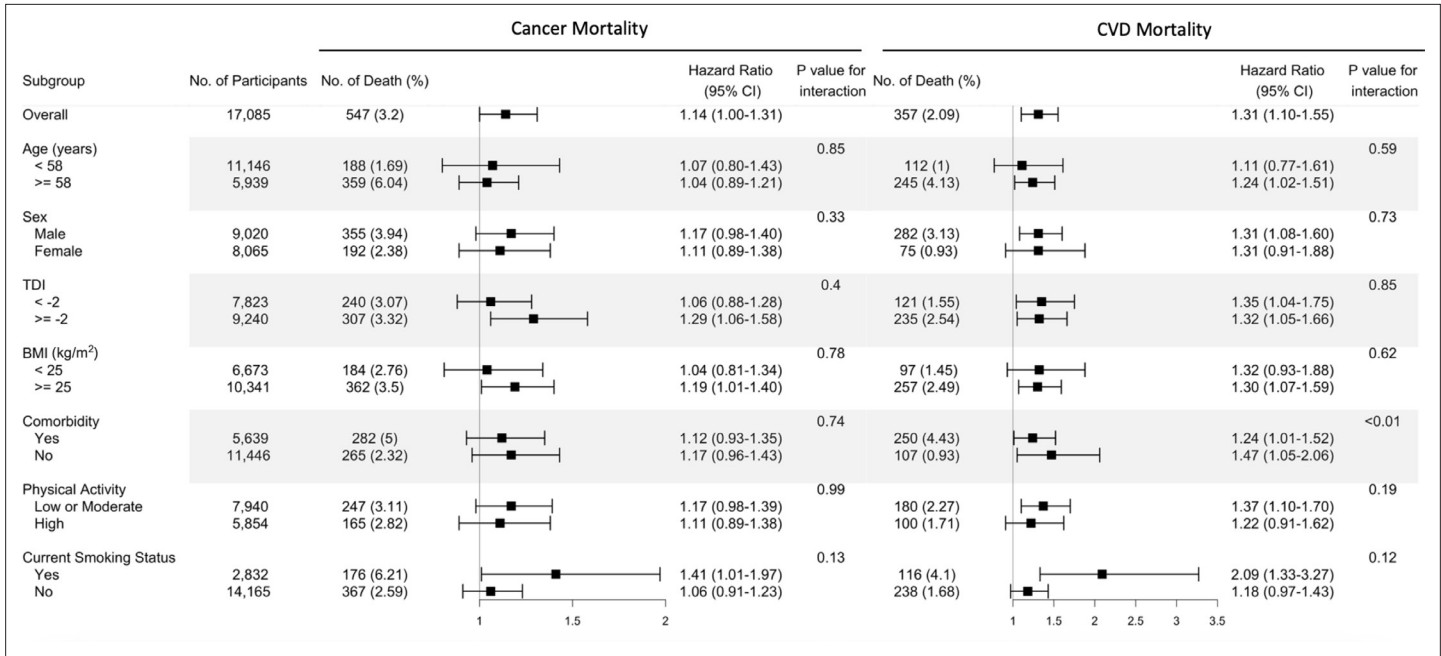

**Figure 2.** Risk estimates of cause-specific mortality for the highest compared with the lowest quintile of the ratio of plasma omega-6 to omega-3 PUFAs, stratified by potential risk factors. Results were adjusted for age (years; continuous), sex (male, female), race (White, Black, Asian, Others), Townsend deprivation index (continuous), assessment centre, BMI (kg/m²; continuous), smoking status (never, previous, current), alcohol intake status (never, previous, current), physical activity (low, moderate, high), and comorbidities (yes, no).

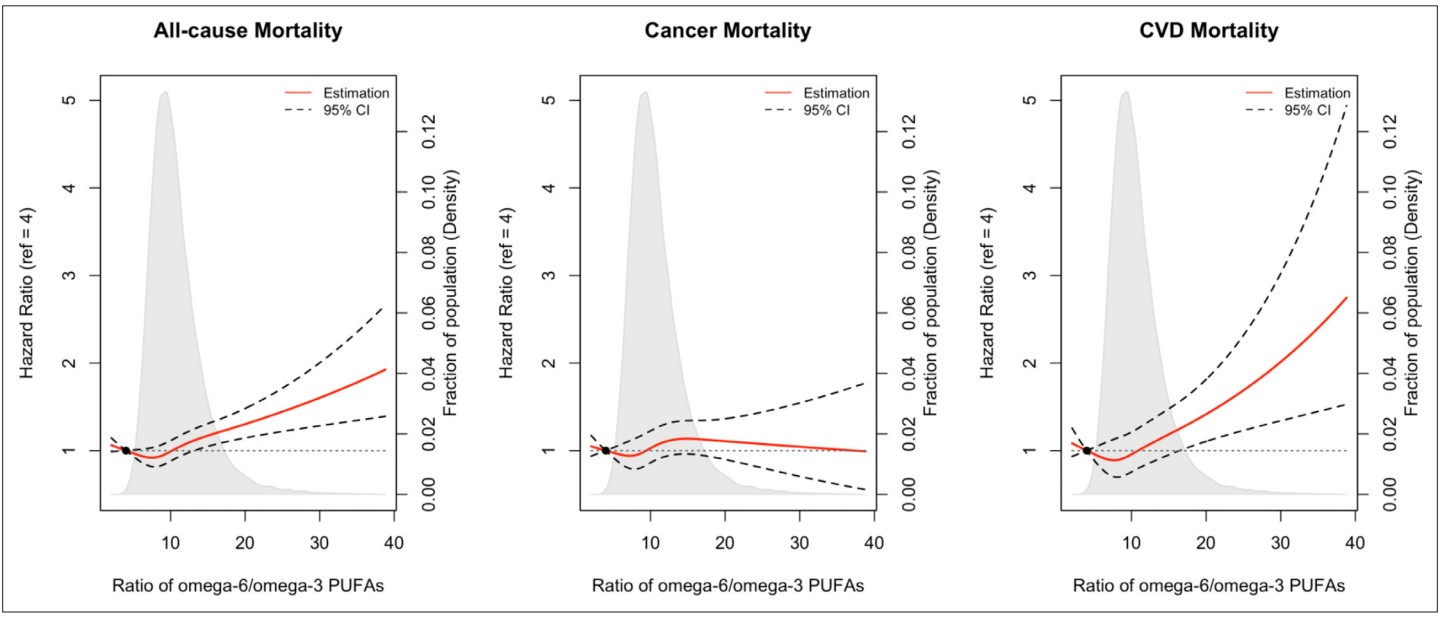

**Figure 3.** Associations of the ratio of omega-6/omega-3 PUFAs with all-cause and cause-specific mortality evaluated using restricted cubic splines. Hazard ratios and omega ratios are presented in the vertical and horizontal axis, respectively. The best estimates and their confidence intervals are presented as solid red lines and dotted black lines, respectively. The ratio 4 was selected as a reference level, and the x-axis depicts the ratio from 0 to 40. Potential nonlinearity was identified for all-cause mortality (p<0.05) and CVD-caused mortality (p<0.05), but not for cancer-caused mortality (p=0.12). All HRs are adjusted for age (years; continuous), sex (male, female), race (White, Black, Asian, Others), Townsend deprivation index (continuous), assessment centre, BMI (kg/m2; continuous), smoking status (never, previous, current), alcohol intake status (never, previous, current), physical activity (low, moderate, high), and comorbidities (yes, no).

under the fully adjusted Model 3. For example, when comparing those in the highest omega-3% quintile to the lowest quintile, the fully adjusted HRs (95% CI) for all-cause, cancer, and CVD mortality were, respectively, 0.69 (0.63, 0.76), 0.75 (0.65, 0.87), and 0.68 (0.57, 0.82) (*Supplementary file 2*: Table S3). The corresponding HRs for the omega-6% were 0.77 (0.70, 0.85), 0.80 (0.68, 0.92), and 0.83 (0.68, 1.02) (*Supplementary file 2*: Table S4). Furthermore, in another joint analysis of the omega-3% and omega-6%, the lowest risk for all-cause and cancer mortality was observed among those in the joint highest categories of the two fatty acids (*Supplementary file 2*: Table S5). For example, when comparing those in the highest quintiles of the two fatty acids to the group with the joint lowest group, the HRs (95% CI) for all-cause and cancer mortality were, respectively, 0.48 (95% CI, 0.35, 0.67) and 0.53 (95% CI, 0.33, 0.86). In the analysis of dietary PUFAs with mortality, the effect sizes were smaller and less significant compared with those of the corresponding plasma levels. When analyzed as continuous variables, dietary omega-3%, omega-6% and omega-6/omega-3 ratio were all significantly associated with all-cause, cancer, and CVD mortality, in directions consistent with their plasma counterparts. The trend test across the five quintiles revealed significant associations between dietary omega-3% and omega-6/omega-3 ratio with cancer mortality (P for trend <0.001 and=0.002, respectively; *Supplementary file 2*: Table S6). The correlation between dietary omega-3% and omega-6% was $r$=0.41 (p<0.01).

## Mediation analysis for intermediate biomarkers

The baseline serum biomarkers of the participants across quintiles of the ratio of omega-6/omega-3 were summarized in *Supplementary file 2*:Table S7. The estimates for the three paths a (association between plasma omega-6/omega-3 ratio and biomarkers), b (association between biomarkers and mortality outcomes), and c (association between plasma omega-6/omega-3 ratio and mortality outcomes), as well as the percentage of indirect effect (a to b) among the overall associations, were reported in *Supplementary file 2*: Table S8 and Figure S1. These analyses identified CRP and SHBG as potential mediators for the association of plasma omega-6/omega-3 ratio with all-cause mortality (proportion of mediating = 4% and 5.0%, respectively).

## Other sensitivity analyses

We performed analyses to examine the associations of DHA% and LA% with mortality outcomes. The correlation between DHA% and LA% was relatively low with $r$=0.03 (p<0.01). Under the fully-adjusted model, both DHA% and LA% were inversely associated with all three mortality outcomes, regardless of being treated as continuous or quintiles (*Supplementary file 2*: Table S9). After further adjustment of the fish oil supplementation status, the associations between the ratio of omega-6/omega-3 and mortality outcomes were slightly attenuated yet did not alter the main findings (*Supplementary file 2*: Table S10). Our primary analysis excluded participants with missing information (i.e. a complete-case analysis). We performed a sensitivity analysis using the multiply-imputed datasets, and there were no substantial changes (*Supplementary file 2*: Table S11). Moreover, the exclusion of participants who died during their first-year follow-up did not materially alter the results (*Supplementary file 2*: Table S12). The baseline characteristics are comparable between participants with or without exposure information (*Supplementary file 2*: Table S13).

## Discussion

In this prospective population-based study of UK individuals, we showed that a higher ratio of plasma omega-6/omega-3 fatty acids was positively associated with the risk of all-cause, cancer, and CVD mortality. These associations were independent of most risk factors examined, including age, sex, TDI, BMI, comorbidities, and physical activity, but they were stronger in current smokers. These relationships were linear for cancer mortality but not for all-cause and CVD mortality. For those two outcomes, the risk of mortality first decreased at lower ratios and then increased, with an inflection point around the ratio of 8. Moreover, omega-3 and omega-6 PUFAs in plasma were consistently and inversely associated with all-cause, cancer, and CVD mortality, with omega-3 showing stronger effects.

To date, studies that examined the relationship between the ratio of omega-6/omega-3 PUFAs and mortality in the general population are sparse (*Harris et al., 2017*; *Otsuka et al., 2019*; *Zhuang et al., 2019*; *Noori et al., 2011*). Similar to our finding, in a 2017 report from a prospective women cohort

study (n=6501; 1875 all-cause deaths, 617 CVD deaths, 462 cancer deaths) (*Harris et al., 2017*), the adjusted HR for all-cause mortality was 1.10 (1.02–1.19) per 1-SD increase of omega-6/omega-3 ratio in red blood cells; however, the effects were not significant for cancer and CVD deaths. Another study supporting our results was conducted on elderly Japanese individuals (n=1054; 422 deaths). It found that the ratio of an omega-3 fatty acid, EPA, to an omega-6 fatty acid, arachidonic acid (ARA), is inversely associated with all-cause mortality, with an HR of 0.71 (95% 0.53–0.96) comparing the highest to the lowest tertile (*Otsuka et al., 2019*). Findings of previous studies based on dietary intake were null (*Zhuang et al., 2019*; *Noori et al., 2011*). A prospective cohort involving 145 hemo-dialysis patients enrolled in Southern California during 2001–2007 (42 all-cause deaths) showed that the estimated HR (95% CI) for all-cause mortality among those in the lowest relative to the highest quartiles of dietary omega-6/omega-3 ratio was 0.37 (0.14–1.08) (*Noori et al., 2011*). Although the effect was not significant, it still shows the protective trend of a lower omega-6/omega-3 ratio against premature death, when taking the sample size and study population into consideration. In a 2019 report based on two population-based prospective cohorts in China (n=14,117, 1007 all-cause deaths) and the US (n=36,032, 4826 all-cause deaths), the effect of the omega-6/omega-3 ratio intake was not significant (HR (95% CI) is 0.95 (0.80–1.14) for China and 0.99 (0.89–1.11) for US, respectively) (*Zhuang et al., 2019*). These discrepancies may be explained by the usage of circulating biomarkers or dietary intakes for calculating the omega-6/omega-3 ratio. Indeed, we performed the same association analyses for dietary PUFAs and found that their associations with mortality have smaller effect sizes and are less statistically significant, although maintaining the same effect directions as plasma PUFAs. The estimated dietary intakes may be inaccurate due to recall bias or outdated food databases (*Shim et al., 2014*). The circulating level is a more objective measurement of PUFA status and thus provides a more reliable picture of the effects of omega-3 and omega-6 PUFAs on mortality. Additionally, we observed that the omega-6/omega-3 ratio tends to have stronger effects on mortality outcomes in current smokers. This is consistent with a previous observation that the association between habitual fish oil supplementation and a lower risk of all-cause mortality is stronger in current smokers (*Li et al., 2020*). Moreover, we found that current smokers in UK Biobank tend to have higher omega-6/omega-3 ratios. Similarly, prior studies reported that current smokers have lower circulating omega-3 PUFAs than non-smokers but similar circulating levels of omega-6 PUFAs (*Murff et al., 2016*; *Scaglia et al., 2016*). It is likely that omega-3 PUFAs play more important roles in current smokers, resulting in a stronger effect of the omega-6/omega-3 ratio on mortality. However, the underlying mechanism awaits future investigations.

A large number of existing observational studies documented the inverse association of circulating levels and intake of omega-3 PUFAs with mortality (*Harris et al., 2018*; *Kleber et al., 2016*; *Zhang et al., 2018*; *Chen et al., 2016*; *Eide et al., 2016*; *Harris et al., 2021*; *Lee et al., 2009*; *Lindberg et al., 2008*; *Pertiwi et al., 2021*), which is in line with our finding that individuals in the highest quintile of the plasma omega-3% had approximately 30% lower risk for all-cause and CVD mortality and 25% lower risk for cancer mortality, when compared to those in the lowest quintile. Our analysis of dietary omega-3% also revealed inverse associations with all-cause and cancer mortality, with 9% and 18% lower risk, respectively. Few previous studies have examined the association in generally healthy populations (*Harris et al., 2018*; *Zhang et al., 2018*; *Chen et al., 2016*; *Harris et al., 2021*). In a 2016 meta-analysis (*Chen et al., 2016*) of seven epidemiologic studies for dietary omega-3 PUFAs intake and four studies for circulating levels, the estimated relative risk for all-cause mortality was 0.91 (95% CI: 0.84–0.98) when comparing the highest to the lowest categories of omega-3 PUFA intake. In two 2018 reports of population-based studies in the US, the circulating level and dietary intake of omega-3 PUFAs were significantly associated with lower total mortality (*Harris et al., 2018*; *Zhang et al., 2018*). In a 2021 meta-analysis of 17 epidemiologic studies, comparing the highest to the lowest quintiles of circulating EPA and DHA, the estimated HRs for all-cause, CVD, and cancer mortality were 0.84 (0.79–0.89), 0.80 (0.73–0.88), and 0.87 (0.78–0.96), respectively (*Harris et al., 2021*). Moreover, inverse associations of omega-3 PUFAs biomarkers with total mortality were found in patients with myocardial infarction (*Lee et al., 2009*; *Pertiwi et al., 2021*), type 2 diabetes (*Harris et al., 2020*), and other diseases (*Kleber et al., 2016*; *Eide et al., 2016*; *Lindberg et al., 2008*; *Pottala et al., 2010*); inverse associations of omega-3 PUFAs with CVD mortality were also reported in these patient groups (*Kleber et al., 2016*; *Harris et al., 2020*; *Pertiwi et al., 2021*). However, there are also reports of a null relationship between omega-3

PUFAs and mortality (*Miura et al., 2016*; *Otsuka et al., 2019*; *Miura et al., 2018*). One possible explanation for the discrepancies could be the limited statistical power due to the small sample size and the small number of events. Moreover, the inconsistency may be due to unique sample characteristics; one study only involved elder patients (*Otsuka et al., 2019*), and one only involved women (*Miura et al., 2018*).

A smaller number of studies have evaluated the associations of omega-6 PUFAs with mortality. Our findings in the large UK prospective cohort are in accordance with the results of several previous studies that increased circulating levels of omega-6 PUFAs were associated with decreased all-cause mortality (*Delgado et al., 2017*; *Wu et al., 2014*; *Zhuang et al., 2019*; *Warensjö et al., 2008*). Moreover, a 2019 meta-analysis involving 18 cohort studies and 12 case-control studies showed that an omega-6 PUFA, linoleic acid, was inversely associated with CVD mortality; the HR per interquintile range was 0.78 (95% CI, 0.70–0.85) (*Marklund et al., 2019*). Other studies, however, did not support such inverse associations (*Harris et al., 2017*; *Harris et al., 2020*; *Miura et al., 2016*; *Otsuka et al., 2019*; *Miura et al., 2018*). Although the findings were inconsistent, no previous studies reported harmful effects of omega-6 PUFAs on mortality (*Harris et al., 2017*; *Wu et al., 2014*; *Harris et al., 2020*; *Miura et al., 2016*; *Otsuka et al., 2019*; *Zhuang et al., 2019*; *Miura et al., 2018*; *Warensjö et al., 2008*; *de Lorgeril et al., 1994*; *Petrone et al., 2013*). Our studies support the inverse associations between omega-6 PUFAs and all-cause and cause-specific mortality. Further research is needed to investigate the health impacts of omega-6 PUFAs in laboratory studies, epidemiological investigations, and clinical trials.

We observed that the omega-6/omega-3 ratio is positively, while both of its numerator and denominator, omega-6% and omega-3%, are negatively associated with mortality. Our findings support that both omega-6 and omega-3 PUFAs are protective against death and that the positive associations of the omega-6/omega-3 ratio with mortality outcomes are likely due to the stronger effects of omega-3 than omega-6 PUFAs. Our mediation analysis with clinical biomarkers for CVD and cancer revealed that CRP and SHBG partially medicated the effects of the omega-6/omega-3 ratio on all-cause mortality. However, the proportions mediated are relatively small (~5%), suggesting the presence of direct effects or unexamined factors. Our findings call for future studies that evaluate the utility and investigate the mechanisms of the omega-6/omega-3 ratio.

Strengths of our current study include the use of objective PUFA biomarkers in plasma instead of the estimated intakes from dietary questionnaires, which increases the accuracy of exposure assessment. Moreover, the prospective population-based study design, large sample size, long duration of follow-up, and detailed information on potential confounding variables, substantially mitigate the possible complications from reverse causality and confounding bias. In several sensitivity analyses, most of the documented associations remain materially unchanged, indicating the robustness of our results.

Several potential limitations deserve attention. First, plasma omega-3 and omega-6 PUFAs were measured only once at baseline. Their levels may vary with diet or other lifestyle factors, which could cause misclassification over follow-up. However, some studies demonstrated that multiple measurements of omega-3 PUFAs have been consistent for a 6-month period (*Kobayashi et al., 2001*). Moreover, the 13 year within-person correlation for circulating omega-6 PUFAs was comparable to such correlations for other major CVD risk factors (*Clarke et al., 1999*) Thus, the single measurement of PUFAs at baseline, although not perfect, provides us with adequate information to investigate the relative long-term effects. Second, although we adjusted for many potential confounders in the model, we cannot rule out the imprecisely measured and unmeasured factors. Third, we acknowledged that the cohort did not mirror the broader UK demographic in terms of socioeconomic and health profiles. Participants in the UK Biobank generally exhibited better health and higher socioeconomic status than the average UK resident, potentially influencing the disease prevalence and incidence rates. Nonetheless, the UK Biobank's extensive sample size and comprehensive exposure data enable the generation of valuable estimates for exposure-disease associations. These estimates have been corroborated by findings from more demographically representative cohorts (*Batty et al., 2020*; *Fry et al., 2017*). Last, although we included individuals of different ancestries in the analysis, over 90% of the participants were of European ancestry. The generalizability of our findings across ancestries requires future verification.

## Conclusions

In this large prospective cohort study, we documented robust positive associations of the plasma omega-6/omega-3 fatty acids ratio with the risk of all-cause, cancer, and CVD mortality. Moreover, we found that plasma omega-3 and omega-6 PUFAs were independently and inversely associated with the three mortality outcomes, with omega-3 fatty acids showing stronger inverse associations. Our findings support the active management of a high circulating level of omega-3 fatty acids and a low omega-6/omega-3 ratio to prevent premature death. Future research is warranted to further test the causality, such as Mendelian randomization and randomized controlled trials. Mechanistic research, including comprehensive mediation analysis, in-depth experimental characterization in animal models or cell lines, and intervention studies, is also needed to unravel the molecular and physiological underpinnings.

## Acknowledgements

This research has been conducted using the UK Biobank Resource under Application Number 48818. This work uses data provided by patients and collected by the NHS as part of their care and support. These data are copyrighted, 2022, NHS England. Reused with the permission of the NHS England and UK Biobank. All rights reserved. This research used data assets made available by National Safe Haven as part of the Data and Connectivity National Core Study, led by Health Data Research UK in partnership with the Office for National Statistics and funded by UK Research and Innovation (grant ref MC_PC_20058). We would like to express our heartfelt gratitude to the UK Biobank participants and administrative staff.

## Additional information

### Funding

| Funder | Grant reference number | Author |
|---|---|---|
| National Institute of General Medical Sciences | R35GM143060 | Kaixiong Ye |

The funders had no role in study design, data collection and interpretation, or the decision to submit the work for publication.

### Author contributions

Yuchen Zhang, Data curation, Formal analysis, Investigation, Visualization, Methodology, Writing - original draft; Yitang Sun, Data curation, Investigation, Writing – review and editing; Qi Yu, Formal analysis, Investigation, Writing – review and editing; Suhang Song, J Thomas Brenna, Investigation, Writing – review and editing; Ye Shen, Conceptualization, Supervision, Investigation, Project administration, Writing – review and editing; Kaixiong Ye, Conceptualization, Resources, Supervision, Funding acquisition, Investigation, Project administration, Writing – review and editing

### Author ORCIDs

Yuchen Zhang ⓘ http://orcid.org/0000-0002-8992-5258
Yitang Sun ⓘ http://orcid.org/0000-0002-6564-8815
Qi Yu ⓘ http://orcid.org/0009-0007-7585-1183
Suhang Song ⓘ http://orcid.org/0000-0003-1934-632X
J Thomas Brenna ⓘ https://orcid.org/0000-0001-9494-4245
Ye Shen ⓘ http://orcid.org/0000-0002-6662-0048
Kaixiong Ye ⓘ http://orcid.org/0000-0003-4658-7292

### Ethics

Human subjects: The UK Biobank received ethical approval from the research ethics committee (reference ID: 11/ NW/0382). Written informed consent was obtained from participants.

Reviewer #3 (Public review) https://doi.org/10.7554/eLife.90132.3.sa1

Author response https://doi.org/10.7554/eLife.90132.3.sa2

## Additional files

### Supplementary files
- MDAR checklist
- Supplementary file 1. Supplementary Table 1 for literature review.
- Supplementary file 2. Supplementary Tables 2 - 14 for additional results.
- Supplementary file 3. Directed acyclic graph to explain mediation.
- Reporting standard 1. STROBE checklist.

### Data availability
The datasets analyzed during the current study are available from the UK Biobank through an application process (http://www.ukbiobank.ac.uk/). R scripts were deposited on GitHub (copy archived at *Zhang, 2024*).

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
