## [Editor Report · eLife assessment]

The manuscript provides **convincing** evidence that both circulating omega-6 and omega-3 PUFAs are associated with lower all-cause, cancer, and cardiovascular mortality in the UK BioBank population and that omega-3s have a stronger effect than omega-6s. The findings have **important** public health implications.

---

## [Referee Report · Reviewer #3 (Public review)]

Summary:

The authors are trying to find out whether the levels of omega-6 and omega-3 fatty acids in the blood are linked to the likelihood of dying from anything, of dying from cancer and of dying from cardiovascular disease. They use a large dataset called UKBiobank where fatty acid levels were measured in blood at the start of the study and what happened to the participants over the following years (average of 12.7 years) was followed. They find that both omega-6 AND omega-3 fatty acids were linked with less likelihood of dying from anything, from cancer and from cardiovascular disease. The effects of omega-3s were stronger. They then made a ratio of omega-6 to omega-3 fatty acids and found that as that ratio increased risk of dying also increased. This supports the idea that omega-3s have stronger effects than omega-6s.

Strengths:

This is a large study (over 85,000 participants) with a good follow up period (average 12.7 years). Using blood levels of fatty acids is superior to using estimated dietary intakes. The authors take account of many variables that could interfere with the findings (confounding variables) - they do this using statistical methods.

Weaknesses:

UKBioBank is not entirely representative of the UK population.

---

## [Author Response]

The following is the authors’ response to the original reviews.

**Public reviews:**

**Reviewer #1 (Public review):**
Summary:The manuscript offers a commendable exploration into the relationship between plasma omega-6/omega-3 fatty acid ratios and mortality outcomes.Strengths:The chosen study design and analytical techniques align well with the research objectives, and the results resonate with existing literature.Weaknesses:Lack of information on the selection criteria for participants; 5. The analysis of individual PUFAs is not appropriate; The definition of comorbidities is vague; The rationale of conducting the mediation analysis of blood biomarkers is not given.

Thank you for your insightful feedback and for acknowledging the strengths of our manuscript, particularly regarding the alignment of our study design and analytical methods with our research objectives. Your recognition of how our results resonate with existing literature is greatly appreciated.

Addressing the concerns you've raised:

Selection Criteria for Participants: In the “Methods-Study population” section, we have outlined the exclusion criteria for participant selection. This information provides comprehensive insight into our methodology for selecting the study cohort.

Analysis of Individual PUFAs: We acknowledge your concern regarding the analysis of individual PUFAs due to their inter-correlations in plasma levels. However, the correlations between omega-3% and omega-6% (r = -0.12) and between DHA% and LA% (r = 0.03) are actually low. Because DHA is one of omega-3 PUFAs, we did not include PUFAs in the same model. Similar considerations apply to LA and omega-6. We believe that exploring the effects of individual fatty acids adds valuable depth to our research. Both DHA and LA have been included in the same model due to their low correlation, with careful adjustments for confounding factors to provide a nuanced understanding of their individual impacts on mortality.

Definition of Comorbidities: The definition of comorbidities, including hypertension, diabetes, and longstanding illness, is elaborated under the Methods section. These conditions were identified through self-reported data collected via the Assessment Centre Environment (ACE) touchscreen questionnaire, allowing us to capture a broad range of chronic conditions as reported by participants.

Rationale for Mediation Analysis: Initially, our approach to mediation analysis included various blood biomarkers available in the UK Biobank database to explore the potential underlying pathways. However, upon considering your feedback regarding the overlap of fatty acids with lipid classes or lipid particles in plasma, we have decided to remove these elements from our mediation analysis.

**Reviewer #2 (Public review):**
Summary:This study utilized a large sample from the UK Biobank which enhanced statistical robustness, employed a prospective design to establish clear temporal relationships, used objective biomarkers for assessing plasma omega-6/omega-3 ratio, and investigated various mortality causes including CVD and cancer for a holistic health understanding.Strengths:The authors used a large sample size, employed a prospective design, and investigated various mortality.Weaknesses:Analyzing n-3 and n-6 PUFAs separately might be less instructive. It might not be methodologically sound to treat TG, HDL, LDL, and apolipoproteins as mediators. It's imperative to exercise caution when drawing causal conclusions from the observed correlations. The manuscript might propose potential research trajectories.

We are grateful for your thoughtful analysis of our study's strengths and for your constructive feedback on areas for improvement.

Response to Weaknesses:

Analyzing n-3 and n-6 PUFAs Separately: We recognize the challenge in analyzing n-3 and n-6 PUFAs separately due to their correlations. However, the correlation between n-3% and n-6% in UK Biobank was actually relatively low (r = -0.12). We include them in one model to test if both are associated with the outcomes after controlling for the effects of the other. Indeed, both were negatively associated with the mortality outcomes in our analysis. We believe our supplemental analysis of n-3 and n-6 PUFAs provides useful information to the readers, in addition to our findings based on the n-6/n-3 ratio.

Mediation Analysis of TG, HDL, LDL, and Apolipoproteins: We appreciate your insight on the methodological considerations of treating these biomarkers as mediators. After careful review and in line with suggestions from another reviewer, we have removed these elements from our mediation analysis. This revision improves the net scientific rigor of our work, ensuring that our conclusions are drawn from the most robust and methodologically sound of our analyses.

Causal Conclusions from Correlations: We fully agree with the need for caution in interpreting correlations in observational studies. To this end, we have avoided implying causality in our manuscript. Terms suggesting causality, like "protective effects," have been replaced with "inverse associations" to more accurately represent our findings. This adjustment enhances the clarity and accuracy of our conclusions.

Proposing Future Research Trajectories: Recognizing the importance of advancing causal and mechanistic understanding in this field, we have called for future studies to further examine causality and characterize molecular mechanisms of the observed associations in our study.

**Reviewer #3 (Public review):**
Summary:The authors are trying to find out whether the levels of omega-6 and omega-3 fatty acids in the blood are linked to the likelihood of dying from anything, of dying from cancer and of dying from cardiovascular disease. They use a large dataset called UK Biobank where fatty acid levels were measured in blood at the start of the study and what happened to the participants over the following years (average of 12.7 years) was followed. They find that both omega-6 AND omega-3 fatty acids were linked with less likelihood of dying from anything, from cancer and from cardiovascular disease. The effects of omega-3s were stronger. They then made a ratio of omega-6 to omega-3 fatty acids and found that as that ratio increased risk of dying also increased,. This supports the idea that omega-3s have stronger effects than omega-6s.Strengths:This is a large study (over 85,000 participants) with a good follow up period (average 12.7 years). Using blood levels of fatty acids is superior to using estimated dietary intakes. The authors take account of many variables that could interfere with the findings (confounding variables) - they do this using statistical methods.Weaknesses:There are several omega-6 and omega-3 fatty acids - it is not clear which ones were actually measured in this study

Thank you for recognizing the strengths of our study, including the large sample size, the duration of follow-up, and our methodological approach to using blood levels of fatty acids and addressing potential confounders.Regarding the weakness you've highlighted, we understand the importance of specifying which omega-6 and omega-3 fatty acids were analyzed in our study. We have revised the Method section to provide detailed information about how the exposures were measured.

**Recommendations for the authors:**

**Reviewer #1 (Recommendations for the authors):**
To elevate the manuscript's scholarly rigor, I propose the following refinements:(1) The manuscript lacks information on the selection criteria for participants and the representativeness of the UK Biobank cohort. It is important to provide details on how participants were selected and whether it is representative of the general population, which is crucial for assessing the generalizability of the findings.

We appreciate the opportunity to clarify the participant selection criteria and the representativeness of the UK Biobank cohort within our manuscript. In the “Methods-Study population” section, we delineated the exclusion criteria: "Participants with cancer (n=37,736) or CVD (n=100,972), those who withdrew from the study (n=879), and those with incomplete data on the plasma omega-6/omega-3 ratio (n=277,372) were excluded from this study, leaving 85,425 participants, 6,461 died during follow-up, including 2,794 from cancer and 1,668 from CVD." To further address representativeness, we performed a sensitivity analysis, examining the baseline characteristics of participants included in our study relative to those omitted due to lack of exposure information. This analysis, presented in Additional file 2: Table S13, indicates comparable baseline characteristics across both participant groups, bolstering confidence in the representativeness of our study sample with the general UK Biobank participants.

Regarding the UK Biobank's representativeness with the general population, we acknowledge that the cohort does not mirror the broader UK demographic in terms of socioeconomic and health profiles. Participants in the UK Biobank generally exhibit better health and higher socioeconomic status than the average UK resident, potentially influencing the disease prevalence and incidence rates. Nonetheless, the UK Biobank's extensive sample size and comprehensive exposure data enable the generation of valid estimates for exposure-disease associations. These estimates have been corroborated by findings from more demographically representative cohorts, as highlighted in the studies by Batty et al., and Fry et al..

We recognize the importance of this aspect and will incorporate a discussion on the implications of these factors for the generalizability of our findings in the “Discussion-Limitations” section of our manuscript. We are grateful for this insightful comment and believe that this addition will enhance the manuscript's contribution to the field.

Here is what we added in the “Discussion-Limitations” section of our manuscript: “Third, we acknowledged that the cohort did not mirror the broader UK demographic in terms of socioeconomic and health profiles. Participants in the UK Biobank generally exhibited better health and higher socioeconomic status than the average UK resident, potentially influencing the disease prevalence and incidence rates. Nonetheless, the UK Biobank's extensive sample size and comprehensive exposure data enable the generation of valid estimates for exposure-disease associations. These estimates have been corroborated by findings from more demographically representative cohorts47,48.”

References:

Batty, G. D., Gale, C. R., Kivimäki, M., et al. Comparison of risk factor associations in UK Biobank against representative, general population based studies with conventional response rates: prospective cohort study and individual participant meta-analysis. BMJ. 2020; 368: m131.

Fry A, Littlejohns TJ, Sudlow C, et al. Comparison of Sociodemographic and Health-Related Characteristics of UK Biobank Participants With Those of the General Population. Am J Epidemiol. 2017;186(9):1026–34.

(2) The study sample included different ancestries which may introduce confounding from genetic background. As over 90% of the participants were of European ancestry, I recommend excluding individuals of non-European ancestry in the main analysis.

Thank you for raising the concern regarding the inclusion of different ancestries in our study sample and the potential confounding. In our research, we have adhered to the widely accepted practice of including all participants in the study to ensure a comprehensive analysis. Recognizing the predominance of European ancestry within our cohort, which exceeds 90%, we have proactively incorporated ethnicity as a covariate in our statistical models to mitigate confounding influences.

We also considered the feasibility of conducting a stratified analysis for non-European participants. However, the small sample sizes of non-European subgroups do not provide sufficient statistical power to yield reliable or meaningful separate analyses. Consequently, to maintain the integrity and robustness of our findings, we opted to include all participants in the main analysis, adjusting for ethnicity to account for potential confounders.

(3) I noted that a large proportion of participants were excluded due to the lack of data on plasma PUFAs. Were the characteristics of these participants similar to the current analysis sample?

Thank you for raising this very important point. According to UK Biobank, “The EDTA plasma samples were picked randomly and are therefore representative of the 502,543 participants in the full cohort.” (As detailed in Julkunen et al.) Moreover, as noted in our reply to comment #1 above, we performed a sensitivity analysis, examining the baseline characteristics of participants included in our study relative to those omitted due to lack of exposure information.

The results of this analysis are detailed in Additional file 2: Table S13. They demonstrate that the baseline characteristics—such as age, gender, ethnicity, socioeconomic status, and lifestyle habits—are indeed similar between the two groups. This similarity supports the representativeness of our analysis sample and suggests that the exclusion of participants without plasma PUFA data does not introduce a bias that would undermine the validity of our study's findings.

References:

Julkunen H, Cichońska A, Tiainen M, et al. Atlas of plasma NMR biomarkers for health and disease in 118,461 individuals from the UK Biobank. Nat Commun. 2023 Feb 3;14(1):604. doi: 10.1038/s41467-023-36231-7.

(4) The methods section should include a detailed description of the measurement of plasma omega-6/omega-3 fatty acid ratio. It is important to provide information on the analytical techniques used and any quality control measures implemented to ensure the accuracy and reliability of the measurements. Importantly, were repeated measurements done?

Thank you for raising this important point. The details of the metabolomic profiling have been described in previous UK Biobank publications. In this revision, we added a brief description of the measurement process and provided references to previous publications.

Here is what we added in the “Methods- Ascertainment of exposure” section of our manuscript: “Metabolomic profiling of plasma samples was performed with high-throughput nuclear magnetic resonance (NMR) spectroscopy. At the time of this analysis (15 Mar 2023), UK Biobank released the Phase 1 metabolomic dataset, which covered a random selection of 118,461 plasma samples from the baseline recruitment. These samples were collected between 2007 and 2010 and had been stored in −80 °C freezers, while the NMR measurements took place between 2019 and 2020. Detailed descriptions could be found in previous publications about plasma sample preparation, NMR spectroscopy setup, quality control protocols, correction for sample dilution, verification with duplicate samples and internal controls, and comparisons with independent measurements from clinical chemistry assays20-22.”

(5) The analysis of individual PUFAs is not appropriate because plasma levels of these PUFAs, including n-3 PUFAs and n-6 PUFAs, EPA, DHA and AA, are usually correlated. It is hard to differentiate these correlated FAs in Cox model. Whereas the ratio of n-6/n-3 is indeed more comprehensive, and the current analysis demonstrated this ratio as a good marker of mortality. Therefore, the analyses of individual PUFAs can be removed and only focus on the ratio of n-6/n-3.

We resonate with the Reviewer regarding the importance of focusing on the ratio of n-6/n-3. Indeed, the ratio is our focus in this manuscript. We also acknowledge the Reviewer's concern regarding the inclusion of correlated covariates in one statistical model. In that specific analysis, the correlations between omega-3% and omega-6% (r = -0.12) and between DHA% and LA% (r = 0.03) are relatively low. Additionally, we also checked the model for multicollinearity and found that the variance inflation factors (VIFs) were within acceptable ranges. In the fully adjusted model that included omega-3% and omega-6%, all variables had VIFs below 1.13, with omega-3% at a VIF of 1.06 and omega-6% at a VIF of 1.12. Similarly, in the model including DHA% and LA%, all variables also exhibited VIFs under 1.13, with DHA% recording a VIF of 1.07 and LA% a VIF of 1.10. Because DHA is one of omega-3 PUFAs, we did not include them in the same model. We did not include LA and omega-6 in the same model, either. Because the ratio has two components and each component is the sum of multiple individual PUFAs, it is natural to ask which component is more important (e.g., omega-6 or omega-3?), which specific fatty acid is driving the effect of omega-3 PUFAs (e.g., ALA? Or the marine omega-3, EPA and DHA?). We received such feedback frequently when we presented our research previously. Therefore, as an effort to address them, we performed analysis of omega-3, omega-6, DHA, and LA. While we understand the complexities involved in differentiating the effects of individual fatty acids in a Cox model, we believe there is intrinsic value in exploring these relationships further. In our analysis, we have attempted to investigate the effects of individual PUFAs on mortality by including both DHA and LA within the same model due to their low correlation, making adjustments to account for confounding factors (As detailed in Additional file 2: Table S9). Our findings indicate significant inverse associations between both DHA and LA with all-cause, cancer, and cardiovascular disease (CVD) mortality. We agree with the Reviewer that the focus of our manuscript should be the ratio, but also hope the Reviewer will agree with us that keeping the results from individual PUFAs will provide additional useful information to the readers.

(6) The definition of comorbidities (including hypertension, diabetes, and longstanding illness) is vague. Please clarify what diseases longstanding illness includes.

We appreciate the request for clarification regarding the definition of comorbidities in our study, including the categorization of longstanding illness. The information regarding longstanding illnesses was obtained via the Assessment Centre Environment (ACE) touchscreen questionnaire. Participants were asked, "Do you have any long-standing illness, disability, or infirmity?" with the response options being “Yes,” “No,” “Do not know,” and “Prefer not to answer.” For the purposes of our analysis, participants who selected “Yes” were categorized as having a longstanding illness, while the remaining options were grouped as not having a longstanding illness.

This method of classification aligns with our detailed explanation in the “Methods-Ascertainment of covariates” section of the manuscript, where we state that “Comorbidities, including hypertension, diabetes, and longstanding illness, were self-reported at baseline. Longstanding illness refers to any long-standing illness, disability, or infirmity, without other specific information.” It is important to note that this approach is consistent with established precedents in the field. Specifically, the paper by Li et al. in the BMJ utilized a similar definition for comorbidities, reinforcing the validity of our methodology.

References:

Li ZH, Zhong WF, Liu S, et al. Associations of habitual fish oil supplementation with cardiovascular outcomes and all cause mortality: evidence from a large population based cohort study. BMJ. 2020 Mar 4;368:m456.

(7) The rationale of conducting the mediation analysis of blood biomarkers is not given. Since fatty acids can be formed as TG or bound with apolipoproteins in plasma, there is a large overlap of FAs with these biomarkers and thus it is not appropriate to analyze TG, HDL, LDL, and apolipoproteins as mediators.

We are grateful for the insightful feedback regarding the mediation analysis of blood biomarkers. Our mediation analysis aimed to explore the possible biomarkers and biological processes that explain the effects of PUFAs on mortality. Upon reflection, we recognize the complexities introduced by the inherent overlap of fatty acids with different lipid particles and lipid classes in plasma. Considering the potential confounding this overlap presents, and in agreement with your recommendation, we have decided to remove the mediation analyses involving cholesterol, TG, HDL-C, LDL-C, Lp(a), ApoA, and ApoB from our study. We appreciate your guidance on this matter and have updated our manuscript accordingly to reflect these changes.

**Reviewer #2 (Recommendations for the authors):**
(1) Analyzing n-3 and n-6 PUFAs separately might be less instructive given the inherent correlations among plasma levels of n-3 PUFAs and n-6 PUFAs. Also, some important specific PUFAs, such as ALA, AA, EPA, etc. were not available in the UK Biobank data though the authors tried to analyze LA and DHA. The n-6/n-3 ratio, as evidenced by the current analysis, offers a more holistic perspective and might be a superior mortality marker. Thus, I recommend shifting the focus solely to this ratio.

Thank you for the thoughtful comment. Reviewer #1 raised a similar point (comment #5 above). We are glad that both reviewers recognized the importance of the omega-6/omega-3 ratio and agreed with us that the ratio should be the focus of the paper. Please also see our more detailed response above. Briefly, our manuscript centered on the ratio, while the supplemental analysis of omega-3%, omega-6%, DHA%, and LA% provided additional useful information. We included omega-3% and omega-6% in the same model because their correlation was relatively low (r = -0.12). We also checked the model for multicollinearity and found that the variance inflation factors (VIFs) for n-3 PUFAs and n-6 PUFAs were within acceptable ranges. In the fully adjusted model that included omega-3% and omega-6%, all variables had VIFs below 1.13, with omega-3% at a VIF of 1.06 and omega-6% at a VIF of 1.12. Similarly, in the model including DHA% and LA%, all variables also exhibited VIFs under 1.13, with DHA% recording a VIF of 1.07 and LA% a VIF of 1.10. Therefore, we decided to keep the content for omega-3 and omega-6 PUFAs. We hope that Reviewer will agree with us that this content only provides additional information to the readers.

(2) It might not be methodologically sound to treat TG, HDL, LDL, and apolipoproteins as mediators. Since the model included comorbidities as covariates, hypercholesteremia and hypertriglyceridemia seemed to have been adjusted in the analysis. Thus, further adjusting these blood biomarkers for mediation analysis which overlapped with comorbidities is redundant.

We appreciate your critical evaluation of our methodological approach. Your point is well-taken, especially in light of the fact that comorbidities such as hypercholesterolemia and hypertriglyceridemia have been accounted for as covariates in our model. This overlap, as you correctly identified, could indeed render the mediation analysis redundant. In concordance with your recommendation, and incorporating the comments of another reviewer, we have now omitted the mediation analysis involving these blood biomarkers from our study. We believe this adjustment strengthens the methodological soundness of our research and are thankful for your contribution to this refinement. We have updated our manuscript to reflect these changes and ensure our analysis remains robust and free from redundancy.

(3) It's imperative to exercise caution when drawing causal conclusions from the observed correlations. The inherent constraints of observational studies, coupled with potential residual confounding or reverse causality, should be acknowledged.

We concur with the caution against implying causality from correlations observed in our study. As such, we have carefully refrained from claiming any causal relationships within our paper. We acknowledge that the term "protective effects" could suggest a causal inference, and we have revised our language to describe these observations as "inverse associations" to more accurately reflect the nature of our findings.

We have also addressed the inherent limitations of observational research in the Discussion section under 'limitations' of our manuscript. There, we recognize that while we have accounted for many confounders, the possibility of residual confounding cannot be entirely excluded. We also agree that reverse causality is a concern in observational studies. To mitigate this, we performed a sensitivity analysis excluding participants who died within the first year of follow-up. The results from this analysis, which are provided in Additional file 2: Table S12, show consistency with our main findings, suggesting that the observed associations are less likely to be predominantly driven by reverse causation. We are grateful for your insights, which have guided us in strengthening our manuscript and ensuring that our conclusions are presented with the appropriate scientific rigor.

(4) To guide subsequent scholarly endeavors, the manuscript might propose potential research trajectories, such as spearheading randomized controlled trials to delve deeper into the causal nexus between plasma omega-6/omega-3 ratios and mortality outcomes or probing the mechanistic underpinnings of the observed correlations.

We agree that conducting randomized controlled trials could illuminate the potential causal relationships between plasma PUFA biomarkers and mortality outcomes. While the primary focus of our manuscript is to report on associations, we acknowledge the importance of causal analysis in advancing the field. In our secondary analysis, we touched upon mediation effects of blood biomarkers, which could serve as a preliminary step towards establishing causality. Although our current work did not delve deeply into causal mechanisms, the results we have presented may indeed stimulate further exploration. By reporting our mediation analysis results, we aim to provide a foundation that other researchers might build upon. We hope that our work will act as a catalyst for more in-depth studies, such as RCTs or mechanistic investigations, to pursue the questions we have begun to explore.

Following this recommendation, we have revised our Conclusion paragraph and added: “Our findings support the active management of a high circulating level of omega-3 fatty acids and a low omega-6/omega-3 ratio to prevent premature death. Future research is warranted to further test the causality, such as Mendelian randomization and randomized controlled trials. Mechanistic research, including comprehensive mediation analysis, in-depth experimental characterization in animal models or cell lines, and intervention studies, is also needed to unravel the molecular and physiological underpinnings.”

**Reviewer #3 (Recommendations for the Authors):**
(1) Line 32. Delete "a balanced" because a balanced o6:o3 cannot be defined.

Thank you for pointing out the issue with the term "a balanced". Most authors agree with your observation that defining what constitutes a 'balanced' ratio can be ambiguous and potentially misleading. One author, JTB, disagrees that “balance” as a concept is unacceptably ambiguous or misleading. In response, we have removed the words from our manuscript.

(2) In the abstract you should present the findings for omega-6 and omega-3 PUFAs first and then the findings for the ratio.

We appreciate your suggestion to present the findings for omega-6 and omega-3 PUFAs prior to those for the ratio in the abstract. As laid out in the Background section, the ratio was our primary exposure of interest. So, we organized our manuscript by centering on the ratio. We are glad that both Reviewer #1 and #2 expressed a particular interest in the ratio findings and urged us to keep the ratio as the focus. We believe that this emphasis reflects the novel aspects of our research and aligns with the thematic structure of our manuscript.

(3) Line 80. controversial should read uncertain.

Thank you for the suggestion. We have changed “controversial” to “uncertain”.

(4) It is unclear which fatty acids are included in total PUFAs, omega-6 PUFAs and omega-3 PUFAs. It is vital that this is specified.

Thank you very much for your suggestion. We agree that it is important to clarify the specific fatty acids included in the analysis. In the revised manuscript, we emphasized that we analyzed “total omega-6 PUFAs” and “total omega-3 PUFAs”, while “LA is one type of omega-6 PUFAs” and “DHA is one type of omega-3 PUFAs”. We also revised the Method section of “Ascertainment of exposure” to provide more information about how the exposures were measured. Here is what we added in the “Methods- Ascertainment of exposure” section of our manuscript: “Five PUFAs-related biomarkers were directly measured in absolute concentration units (mmol/L), including total PUFAs, total omega-3 PUFAs, total omega-6 PUFAs, docosahexaenoic acid (DHA), and linoleic acid (LA). Of note, DHA is one type of omega-3 PUFAs, and LA is one type of omega-6 PUFAs. Our primary exposure of interest, the omega-6/omega-3 ratio, was calculated based on their absolute concentrations. We also performed supplemental analysis for four exposures, the percentages of omega-3 PUFAs, omega-6 PUFAs, DHA, and LA in total fatty acids (omega-3%, omega-6%, DHA%, and LA%), which were calculated by dividing their absolute concentrations to that of total fatty acids.”